# Web GIS for Sustainable Education: Towards Natural Disaster Education for High School Students

**Jiaqi Li** [1,2], **Haoming Xia** [1,2,*], **Yaochen Qin** [1,2,*], **Pinde Fu** [3], **Xuan Guo** [1,2], **Rumeng Li** [1,2] **and Xiaoyang Zhao** [1,2]

1   National Demonstration Center for Environment and Planning, Henan University, Kaifeng 475001, China; nd990212@henu.edu.cn (J.L.); 104754200150@henu.edu.cn (X.G.); lirm@henu.edu.cn (R.L.); zhaoxy@henu.edu.cn (X.Z.)

2   College of Geography and Environmental Science, Henan University, Kaifeng 475001, China

3   Environmental Systems Research Institute (ESRI), Redlands, CA 92373, USA; pfu@esri.com

\*   Correspondence: xiahm@vip.henu.edu.cn (H.X.); qinyc@henu.edu.cn (Y.Q.); Tel.: +86-135-6950-3779 (H.X.); +86-135-0378-1868 (Y.Q.)

**Abstract:** The rapid development of the web geographic information system (Web GIS) has promoted new vitality in high school geography education, relieved the stress of geography teachers caused by software and technical problems, and made it possible for teachers to devote more energy to geography teaching and research activities. Natural disaster education is not only an important part of the geography curriculum, but also an indispensable aspect of education for sustainable development (ESD) for high school students. The application of Web GIS in the dynamic monitoring, forecast, and early warning of natural disasters is becoming more experienced. Therefore, the application of Web GIS in natural disaster education is quite feasible. How to build a bridge between them is the purpose of this paper. Thus, the paper selects ArcGIS Online, which is not limited by time and space, and analyzes several functions that apply it to geography teaching. These include smart mapping, story maps, 3D web maps, and mobile GIS. Meanwhile, it analyzes the knowledge structure of "natural disasters" in Chinese geography textbooks to guide the subsequent case design. Then, the Web GIS inquiry-based teaching case is formed based on "7.20 Zhengzhou Torrential Rain". It contains knowledge about natural disasters and designs from many aspects, such as the causes, manifestations, and prevention and control of disasters. The discussion identifies a range of specific educational benefits of applying Web GIS to natural disaster education for teachers and schools. Ultimately, it can provide some reference values for geography teachers and other developers to explore curriculum resources and create quality educational models.

**Keywords:** Web GIS; natural disaster education; ESD; ArcGIS Online; geographical core literacy

## 1. Introduction

In 2015, the United Nations Sustainable Development Summit was held in New York; the conference formally adopted the United Nations Sustainable Development Goals (SDGs), which continue to guide global development efforts from 2015 to 2030. In 2016, the Commission on Geographical Education of the International Geographical Union, supported by SDGs and other development principles, promulgated the 2016 International Charter on Geographical Education. The charter articulates the importance of the concept of sustainable development in geography education: citizens who have received geography education can understand the principle of harmonious coexistence between human and environment CGE [1]. Education for sustainable development (ESD) can indeed impact student outcomes in terms of their sustainability consciousness [2].

In 1994, the Chinese government followed the United Nations Conference on Environment and Development and adopted the "China 21st Century Agenda", which emphasized that geography is a critical discipline that contributes to sustainable development [3].

Natural disaster education is an important component of education for sustainable development (ESD). A scientific view of the human–environment relationship by students is the basis for the implementation of ESD. Teachers combine the content system of natural disasters and infiltrate the philosophy of sustainable development into a Geography Fundamentals Course.

The knowledge system of geography involves the whole natural environment and mankind's society. Human–environment interaction is the core perspective of geography research. In 2018, the High School Geography Curriculum Standards (2017) was officially published, mentioning that the geography curriculum aims to equip students with geographical core literacy such as human–environment coordination view, comprehensive thinking, regional cognition, and geographical practice ability. It requires high school students to learn to recognize and appreciate the natural and human environment from a geographical perspective, to understand the harmonious coexistence between human development and the protection of environment, and to have the concept of sustainable development [4]. This shows that geography education bears the burden of education for sustainable development (ESD).

The cultivation of the human–environment coordination view is progressive. Natural disaster education is an important aspect of the cultivation of geographical core literacy, which also is consistent with the need for sustainable social development. It helps students establish a scientific concept of disaster, master the proper methods of disaster reduction and prevention, and safeguard their own life and property safety.

The information barriers created by the rapid development of information technology and the digital divide created by big data in geography have led some geography teachers to choose to remain stagnant, which is not conducive to the development of geography education. It is also an important issue to be solved in the field of geography education to study and explore the "user manual" of geographic information technology suitable for middle school geography teachers of different ages.

The innovation of this paper is to build a bridge between geography education and geographic information technology. Using the interconnection of Web GIS and natural disaster education as an example, it provides geography educators with ideas, methods, and techniques for teaching and research through the analysis of teaching content and technical tools.

## 2. Theoretical Background

### 2.1. Progress on the Integration of Web GIS and Geography Education

Information technology has injected vitality into education. The education of technology and the technologization of education constitute the interactive relationship between technology and education, and it is through this interactive relationship that exogenous technology can be incorporated into the educational process [5]. The rapid development of information technology has made the dissemination of information in the education process cheaper, more flexible, and more widely accessible. It has had an increasingly important impact on the communication between teachers and students.

Web GIS is a pattern, or architectural approach, for implementing a modern GIS. It is powered by web services—standard services that deliver data and capabilities and connect components. Web services are the technical core and an important symbol of modern Web GIS. The strong combination of GIS and Internet makes it possible to run geographic information system wherever there is Internet. Web GIS increases the audience of geographic information, disseminates and simplifies the exchange of geographic data, provides specific structured information, and enables users to access geographic information system applications without using any specific desktop software [6]. Web GIS emerged in 1993 and then quickly stood out for its unparalleled advantages of resource sharing, interoperability, decision support, ease of use, and low cost. Web GIS has been applied in natural disaster monitoring, precipitation monitoring, mapping, environmental protection, urban and rural planning, and regional management. Furthermore, Web GIS is widely used in mega data,

smart city, smart community, artificial intelligence (AI), and other emerging frontier fields, and is playing an increasingly influential role [7]. Web GIS lets geographic information get rid of the fetters of distance, greatly expands the user group of GPS, and makes GIS move from system and science to service.

In the 1970s, the United States began to do research on GIS and develop GIS majors and curriculum in universities. However, at the beginning of the 21st century, awareness and education in GIS was low at all educational levels in the United States [8]. More and more scholars are noticing the necessity of GIS education and conducting research on GIS education at different educational levels (university, high school, middle school) [9]. GIS education, especially in geography, has become so pervasive that it is envisioned as a strategy that can facilitate new ways of teaching, learning, and understanding spatial features [10]. The emphasis of GIS education is not only on how to integrate GIS into geography curricula, but also on the value of GIS in contributing to the realization of curriculum objectives [11].

At the beginning of the 21st century, research on the integration of GIS and education began to emerge and achieved certain results. The main performance is that GIS has been widely used in middle school teaching, and a large number of studies have proven the validity and effectiveness of GIS in K-12 classrooms. Educators also realize the necessity of GIS in geography teaching, many scholars have adopted various methods to promote the real use of GIS during teaching. For example, Favier and van der Schee [12] discussed how to optimize exploratory geography education in GIS. Goldstein and Alibrandi [13] found a correlation between GIS teaching and the increase of standardized test scores through a controlled experiment. Jung [14] introduced a user-centered design (UCD) approach to working with a group of teachers to design and develop GIS learning materials to help teachers effectively learn and implement GIS in the classroom. Grace and Nicola [15] focused their research on how to leverage real-world expertise, where industry experts can influence students' perceptions of GIS's relevance to geography and support their acquisition of geographic knowledge.

Subsequently, Web GIS came into people's vision, and the integration with education became closer and closer. Since the introduction of citizen science networks in the mid-1990s, Web GIS has made an appearance in K-12 Education (K-12: from kindergarten to 12th grade, indicating the number of years of publicly supported elementary and middle school education in the United States). Its representatives include GLOBE, Kan CRN, Journey North, etc. [16]. Their organizations lead the development of Web GIS in the field of education, support interactive websites, and allow students to collect, analyze, and plot their data. Fargher [17] believes that the method of using geographical ability can improve teachers' use of Web GIS and students' ability to think and reason with geographical knowledge and ideas. De Miguel González and De Lázaro Torres [18] made a school digital story atlas based on Web GIS, and analyzed its powerful role in school geography education and its influence on geography teacher education. Bodzin and Anastasio [19] pointed out that GIS maps provide a framework for integrating environmental data across temporal and spatial scales, and using the inquiry education module in Web GIS maps is an ideal choice for earth and environment education. The research of these experts shows that Web GIS, as a unique technical tool in the field of education, has an excellent development trend.

GIS education in China started relatively late Since the 21st century, the attention of educational scholars to geographic information technology has been relatively stable, showing a weak upward trend, but the impact of related research on geographic information technology is generally low. Subsequently, the integration of GIS and geography education became tighter and tighter, and there were more and more relevant studies and papers. Huang et al. [20] analyzed the essential educational functions of GIS in geography teaching the status quo of geography teachers' map capabilities, and put forward an optimization strategy for the sustainable development of GIS geography teaching. This paper focus on natural disaster education in basic geography education, and makes full use of the unique

information advantages of Web GIS to provide more detailed and intuitive reference for other educators.

Nelson Mandela said that "education is the most powerful weapon for changing the world" in one of his classic speeches. This reflects the importance of acquiring knowledge, capabilities, skills, and attitudes in transforming an ever-changing world [21]. As a product of the rapid development of the information technology revolution, Web GIS should be closely integrated with basic geography education. It is not only for high school students to learn the relevant knowledge of Web GIS, but also for geography teachers to develop curriculum resources with effective technical tools. The research on the integration of GIS and geography education has become mature, and it is imperative to seek the strong integration of Web GIS and basic geography education. Web GIS can effectively improve the efficiency of geography teaching, and has been integrated into the new educational tendency. Its success in geographic curriculum has been noticed by many scholars.

### 2.2. Natural Disaster Education in Geography Curriculum

Natural environment is always changeable. When the natural environment changes abnormally and it harms people's lives and property safety, natural disasters come into being. Because of its frequent natural disasters, Japan was one of the earliest countries in the world to offer safety education. Disaster education is the initial stage of safety education, and it is a natural science dedicated to mastering the ability to prevent harm from natural disasters. Shaw et al. [22] analyzed the case of "response to the COVID-19 outbreak in Omuta City, Japan" and concluded that health education and disaster mitigation education are complementary, but there is still a necessity to promote more synergy in order to maximize safety education. At present, the safety problems of primary and middle school students in China present diversified characteristics. The natural environment of China is complex and variable with frequent natural disasters in the world. Natural disasters have randomness and uncertainty, natural disaster education is of great significance to primary and middle school students. In the 5.12 Wenchuan earthquake, all the teachers and students of a middle school in Sichuan province were evacuated in an extremely short period of time. The incident benefited from an in-depth knowledge of disaster preparedness among the school's teachers and students. With frequent emergency evacuation drills, teachers and students mastered the escape route.

Before the new curriculum reform in China, the teaching method of natural disaster education for high school teachers was relatively simple, and mainly relied on a multimedia and lecture method. Teachers do not have the energy and time to design disaster education courses attentively. Therefore, students do not master the disaster prevention and mitigation knowledge properly, and the effectiveness of natural disaster education is weak. The new compulsory geography curriculum of high school has a new chapter on natural disasters. Through the study of this chapter, students can take advantage of theoretical knowledge to explain the causes of common natural disasters, understand measures to avoid and prevent disasters, and become familiar with the application of geographic information technology.

### 2.3. Advantages of Web GIS in Natural Disaster Education

The application of Web GIS in the dynamic monitoring, forecast, and early warning of natural disasters is becoming more experienced. Many scholars have applied it in the monitoring of natural disasters [23], including debris flow, landslide, typhoon, earthquake, tsunami, hurricane, etc. [24–28]. Some scholars have also established an effective EWS (Early Warning System), which can quickly determine the extent of the damage and the damage situation, and provide a basis for formulating disaster mitigation plans and assessing disaster losses. As shown in Figure 1, a week of unprecedented rainfall in Forest Ridge, Colorado, caused floods and mudslides that killed 10 people and caused 2 billion USD in damage in September 2013. Joseph [29] analyzed the areas which are prone to flood and debris flow in Boulder to determine the location of vulnerable people. He used the working

principle of the ArcGIS Online "buffer zone" to predict the areas and impact regions prone to floods and mudslides. The red area represents the geological hazard area, and the blue area represents the potential landslide risk area. We overlaid the "Population of Boulder County" layer and the yellow area is the landslide risk area, which is protected to minimize the loss of life and property if a similar disaster occurs again.

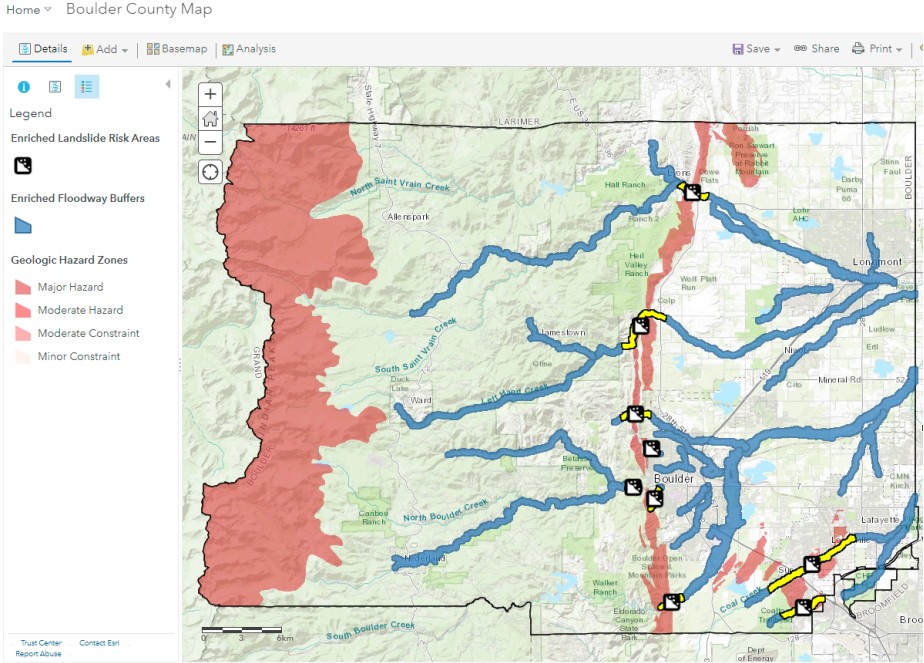

**Figure 1.** Flood and mudslide prone areas in Boulder.

These studies show that GIS tools have a positive effect on students' geography learning. Geography teachers will weaken their interest in geography learning when they try to overcome technology-related difficulties [30]. These projects usually revolve around quantitative research, experiments to derive the effects of technological tools such as GIS on students' geography learning, such as improving their spatial perception [19] and self-efficacy [31]. Teacher training is also a very important part, especially to reduce the technical burden of teachers.

## 3. The Functional Types of Web GIS for High School Geography Teaching

ESRI's ArcGIS Online is a free and opensource GIS with web functions. It has many users and has complete functions, which can meet the needs of geography curriculum resource development and curriculum design.

### 3.1. Smart Mapping

The smart mapping tool of ArcGIS Online Map Viewer can automatically analyze users' data online and provide the most suitable options for their selected maps, analyses, or presentations [17]. Maps and base maps are available via the Web, and the completed atlases create flexible, portable educational modules for science education [19]. This educational module can help geography teachers set layer symbols quickly and intuitively, and easily generate high-quality professional maps without professional cartographic knowledge and software skills.

### 3.2. Story Map

Education is a potentially important application area for story maps. ArcGIS Online provides a web-based story map creation and hosting tool that distributes free accounts to universities to promote the dissemination range of web maps and story maps in education [32]. The designers used the map to organically combine several intrinsically connected

locations with text descriptions and video images, resulting in a dynamic, interesting, and visually insightful story map. Antoniou et al. [33] used the salient features of the methane peninsula as an example, enabling users to interact with maps, text, and images, and to inform professional and non-professional users about specific aspects of the volcanic area. Students are able to zoom in to explore familiar local environments or zoom out to explore larger regional or even global environments, maximizing the usefulness of the interactive maps embedded in the story maps. Students learn by "doing", carry out corresponding operations, and take the initiative to acquire the new geographical knowledge they need, which is conducive to cultivating students' exploratory thinking. For example, Figure 2 shows a story map to enhance people's knowledge of the geographical location of a region while telling a story (https://arcg.is/14Ke8u0, accessed on 20 February 2022).

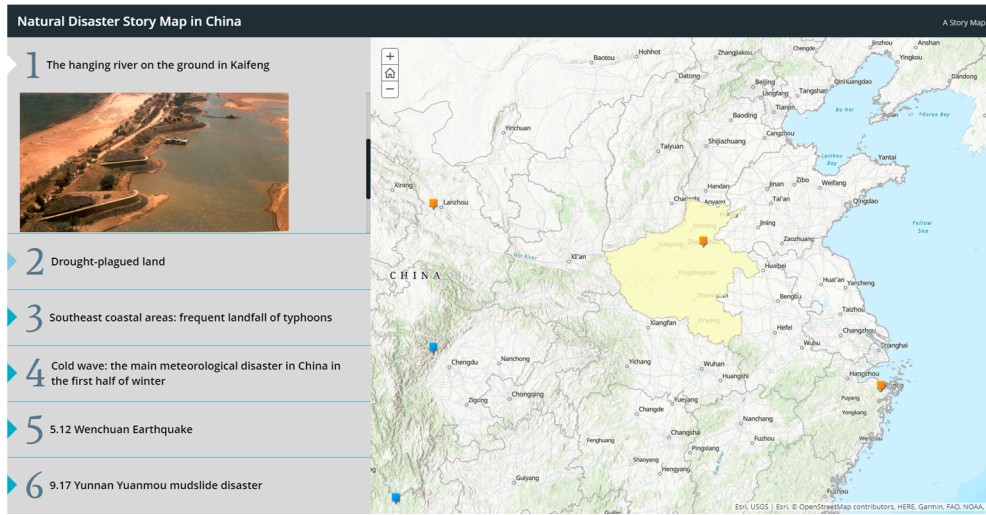

**Figure 2.** Natural disaster story map in China.

### 3.3. 3D Web Maps

Compared with traditional 2D maps, 3D Web maps have obvious advantages in data visualization, spatial analysis, and human–computer interaction. The purpose of any visualization system is to improve users' understanding of the visual information so that users can interact with the visual data and improve their insight [34]. Geospatial technologies such as 3D maps provide a good opportunity to stimulate students' geospatial thinking [35], so the application of 3D maps is indispensable to geographic visualization education.

The "scene" analysis tool of ArcGIS Online is used to view the topography from the perspective of the world. As shown in Figure 3, the western region of North America is mountainous, while the eastern region is relatively flat with high vegetation coverage. Using the profile tool, selecting the desired point can automatically generate the profile line chart and elevation profile data.

### 3.4. Mobile GIS

With the popularization and application of mobile smart devices, the number of users of smartphones and tablets has far exceeded the number of desktop computer users. Meanwhile, mobile communication is developing rapidly, and the number of 5G base stations is also increasing. Mobile GIS has become an important client of Web GIS. Survey 123 for ArcGIS is a forms-centered data collection solution for mobile GIS applications. With its simple interface and easy operation, geography teachers can quickly design questionnaires and share them in the form of QR code or URL. At the same time, geography teachers can analyze the collected questionnaire information on the platform. This paper designed a questionnaire (https://arcg.is/1yqvKn1, accessed on 20 February 2022) to ascertain students' cognition of natural disasters (Figure 4). Teachers analyze the learning

situation based on the questionnaire results to adjust the teaching plan, make up for the shortcomings, and improve their teaching design.

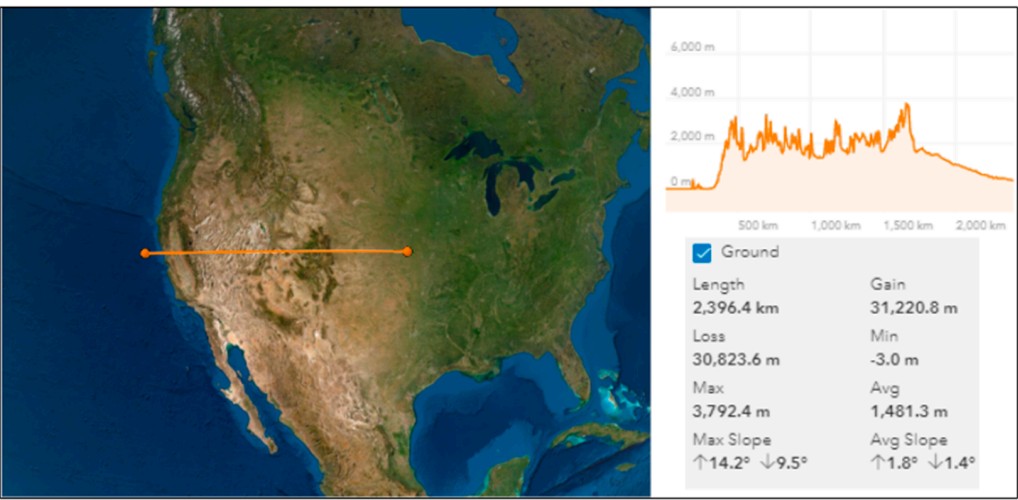

**Figure 3.** ArcGIS Online's "scene" analysis tool—profile.

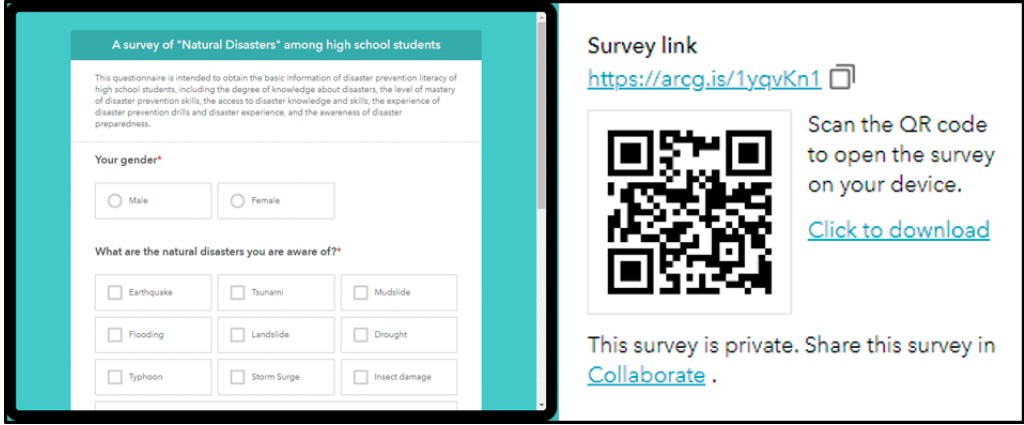

**Figure 4.** A survey of "Natural Disasters" among high school students.

## 4. Knowledge Structure of Compulsory Geography Course "Natural Disaster"

### 4.1. Distribution of Natural Disaster Course Content

With the promulgations of the new curriculum standards for high school geography in China, corresponding high school geography textbooks are also published. Guo et al. [36] pointed out that Chinese middle school textbooks lack sound content and knowledge about sustainable development, and geography teachers lack systematic knowledge support in the process of implementing sustainable development. The fresh geography textbooks emphasize more "ability oriented and lifelong development". Teachers should expand the vision of geography curriculum construction and pay attention to the progress of students' innovation, consciousness, and practical ability. Gong et al. [37] used the method of local individual content analysis. The content related to disaster risk reduction in the geography curriculum standard has undergone continuous changes and finally reached a relatively stable state.

The new geography textbook focuses on "the cultivation of students' geographical core literacy", emphasizes the comprehensive application of geography knowledge points in the content, and makes subtle adjustments in the arrangement order and structure of the textbook. The new textbook devotes entire chapters to the detailed introduction of meteorological and geological hazards and disaster prevention and mitigation (Table 1). It highlights the necessity of understanding the relationship between natural disasters and

human activities and the important role of geographic information technology in disaster prevention and mitigation.

**Table 1.** Distribution of natural disaster knowledge in geography textbooks before and after the new curriculum reform.

|  | Before | After |
|---|---|---|
| **Geography compulsory course** | No introduction to the system: (i)"Weather systems"—meteorological disasters; (ii)"Rational Use of Water Resources"—Drought and Water Scarcity. | Systematic introduction: the whole chapter 6 of Compulsory 1—"Natural disasters". |
| **Geography optional course** | Systematic Explanation: Elective 5—Natural hazards and prevention. | Not mentioned |

### 4.2. Classification of "Natural Disasters"

High school is a critical period for students to construct their critical thinking and logical thinking. Students need to understand the types of natural disasters, analyze the causes of natural disasters, learn the main skills for disaster prevention and mitigation, and cultivate their geographical core literacy. The classification of natural disaster types is important work in natural disaster research and education. According to different standards and principles, the classification of natural disasters also varies widely. Natural disasters include fire storms, dust storms, floods, hurricanes, tornadoes, volcanic eruptions, earthquakes, tsunamis, storms, and other geological processes. In this paper, we introduce and study the classification based on the high school Compulsory 1 Geography textbook (Figure 5).

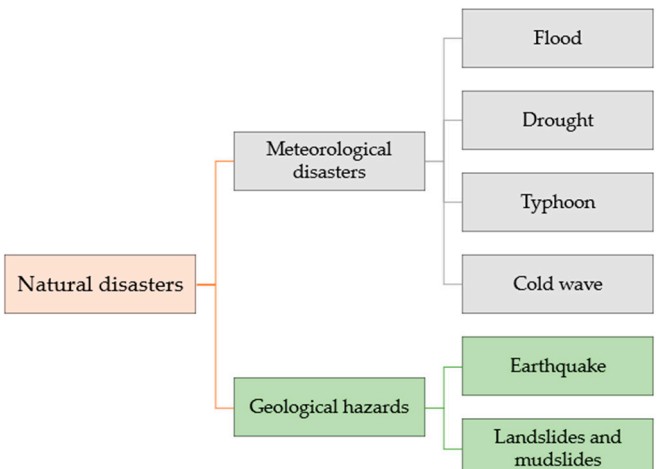

**Figure 5.** Classification of natural disasters.

### 4.3. The Cultivation of Geographical Core Literacy in Natural Disaster Curriculum Content

Under the guidance of the new education philosophy, the setting of geography teaching tasks is student-oriented. It should rely on the real-life situation, take task-driven as the guidance, take group cooperative exploration as the standard, and take the penetration of geographical core literacy as the goal. The following aspects are analyzed in the geographical core literacy of natural disaster education.

It can be analyzed in the following aspects:

- Regional cognition: there are differences in topography and climate in different regions, and the categories of natural disasters are also different. The causes of natural disasters in different regions are investigated and the differences between regions are analyzed.

Meanwhile, it explores how humans and the environment interact with each other and influence the regional ecological environment.

- Comprehensive thinking: the interaction between human society and the natural environment forms the geographical environment, and there are various factors affecting nature. Therefore, students' research and investigation of geography should be comprehensive. Students will develop dialectical thinking by treating a certain type of natural disaster area as a whole, exploring its causes and mechanisms.
- Human–environment coordination view: basic theoretical research of geography is always inseparable from the purpose of the relationship between human and the geographical environment. The knowledge of natural disasters and the cultivation of disaster prevention and mitigation ability are effective ways to cultivate students' human–environment coordination view. Therefore, students could form an attitude of respecting nature and harmonious development.
- Geographical practice ability: teachers guide students to conduct field investigations at natural disaster sites, such as Tangshan Earthquake Memorial Park in Hebei province and Laomao Mountain debris flow site in Liaoning Province. The teacher guides the students to look up the relevant information and collect the local climate characteristics, climate types, historical backgrounds, etc. Teachers are able to work with students to discover, explore, and solve meaningful problems and to develop students' sense of independence and responsibility.

## 5. The Teaching Case Design of Web GIS Applied to High School Students' Natural Disaster Education

Learning geography in life and studying useful geography knowledge are concepts that geography education always carries out. Life is a lesson; disaster is the best textbook. The disaster provides an educational opportunity to enhance students' awareness of disaster prevention and resistance and to improve students' scientific literacy and feelings.

In this paper, "7.20 Zhengzhou Torrential Rain" was introduced into a geography course, which combined theoretical knowledge with life education, science education, and moral education. It is undoubtedly the best first lesson of the fall semester in high school. Based on the mutual penetration of natural disaster education and geographical core literacy, supplemented by major social security events, the teaching ideas of this case were designed (Figure 6).

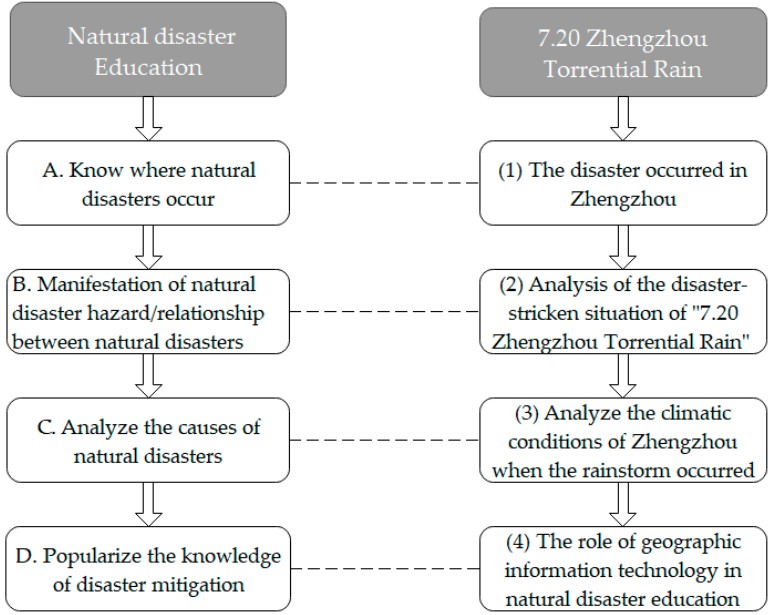

**Figure 6.** "7.20 Zhengzhou Torrential Rain" frame structure.

### 5.1. The Disaster Occurred in Zhengzhou

- Data source: National Earth System Science Data Center
- Technical route:

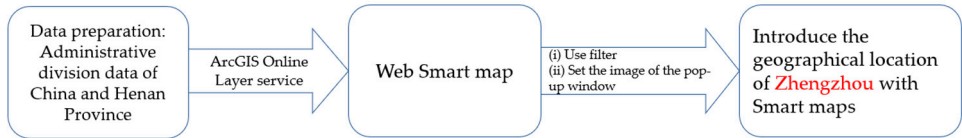

The administrative boundary data of China and Henan Province were obtained from National Earth System Science Data Center (2015). The layer of the Chinese administrative boundary map was firstly added based on the layer service of ArcGIS Online to make an online map. Then, the smart map is shown in Figure 7 using Map Viewer's filters and configuration popup capabilities (https://arcg.is/11S84O, accessed on 20 February 2022). Geography teachers can use the map to guide students to understand the geographical location of Zhengzhou, from large scale to small scale, from shallow to deep.

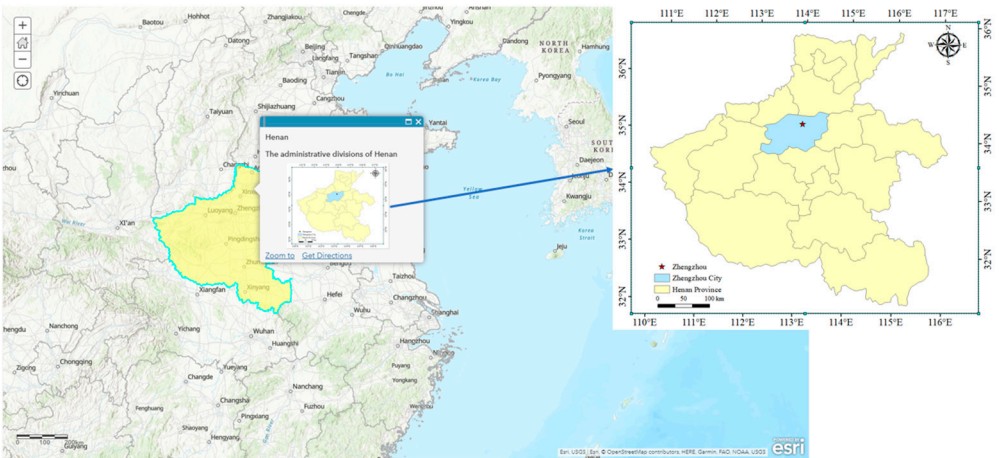

**Figure 7.** The geographical location of Zhengzhou.

- Design Intention

  (1)    Cultivation of regional cognition of high school students.

  Kitchin [38] has shown through research that cognitive maps play an important role in spatial behavior, spatial decision-making, learning and acquisition theories, and real-world applications (such as planning, teaching, and map-making). To study a region in a regional perspective is an essential method of geography learning. Students can understand Zhengzhou from different spatial scales, which makes it easy to create a more profound impression. On this basis, students can develop their regional cognitive ability and establish a correct global outlook and world outlook.

  (2)    Interactive maps can exercise high school students' spatial thinking.

  The design of the Web Smart map can adjust the size of the scale, from the world to see China, from China to see Henan, Henan to see Zhengzhou. Schools equipped with smart classrooms allow teachers to use tablets to deliver instruction. Meanwhile, students can log onto the Internet to analyze and process maps offered by teachers. If you search "Zhengzhou" in the search box, the location of Zhengzhou will be presented in an appropriate proportion, which can get rid of the inherent pattern of reading pictures of students and making them interpret pictures logically. Finally, students' geospatial thinking ability is raised.

*5.2. Analysis of the Disaster-Stricken Situation of "7.20 Zhengzhou Torrential Rain"*

- Case analysis

Material: Henan, located in Huang Huai Hai Plains and the east Asian monsoon climate zone, has suffered from natural disasters since ancient times. According to the historic drought and flood hydrological data, there were 25 severe floods and 21 severe droughts from 2297 BC to 1978 [39]. From 17 July 2021, Henan Province experience a rare extreme heavy rainfall. On 20 July, Zhengzhou was hit by a heavy rainstorm once, with rainfall of 201.9 mm in one hour from 17:00 to 18:00 (Figure 8), setting a new record for hourly rainfall at a national meteorological station in a thousand years (data source: Zhengzhou Meteorological Station). The precipitation lasted for a long time, accumulated rainfall was large, the heavy precipitation region was wide, and the heavy precipitation period was concentrated.

**Figure 8.** Hourly precipitation on July 20 (UTC +8).

Under such rare and extreme rainfall, Zhengzhou suffered from severe flooding, including large-scale urban waterlogging, road collapse, culvert and tunnel flooding, subway and train shutdown, traffic blockage, and citizens being trapped (Figure 9). As the precipitation lasted for a long time and heavy precipitation occurred in a relatively concentrated area, mountain floods in the western and northwestern parts of Henan Province and the risk of geological disasters increased significantly. Different degrees of landslides occurred in the Gongyi City of Zhengzhou, trapping villagers and flooding farmland. Five stations, Songshan, Yanshi, Xinmi, Yichuan, and Dengfeng, broke the historical maximum of daily precipitation since the establishment of the station. In the Yellow River basin and the Huaihe River basin, the water level of reservoirs rose rapidly. The continuous heavy rainfall caused great losses to people's lives and property safety and seriously affected production and life in the province.

As of 12 o'clock on July 23, according to preliminary statistics, 395,989 people were relocated to Zhengzhou, Henan Province, the affected area of crops was 44,209.73 hectares, and the direct economic loss was 65.5 billion yuan. Floods and secondary disasters caused by torrential rain led to 51 deaths [40].

Picture information:

Video materials: CCTV-13 "24 h" directly to Henan flood control and relief, rain intensified, three orange alerts.

(URL: https://tv.cctv.com/2021/07/21/VIDEnaO2SIqSXwqnqniGNJvQ210721.shtml?
spm=C53156045404.P4yTQKr09uhz.0.0, accessed on 20 February 2022)

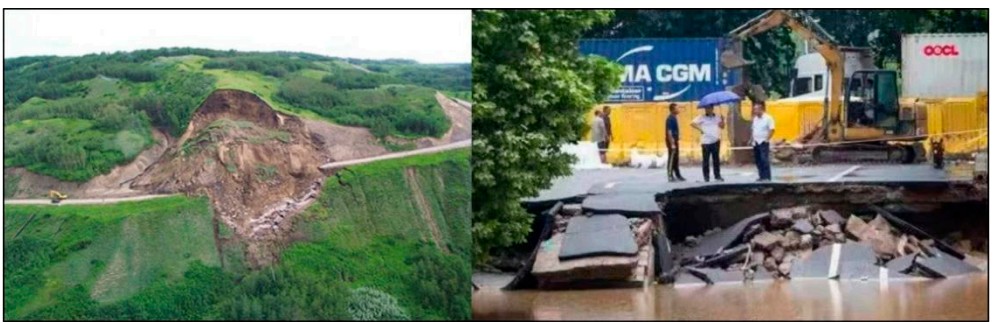

**Figure 9.** Landslides and road collapses.

- Student Inquiry activity:
  (1)     What is the direct cause of floods and landslides?
  (2)     What are the similarities and differences between floods and landslides?
  (3)     Which aspects of human society are threatened by floods and landslides?
- Design Intention
  (1)     Cultivate students' geographical comprehensive thinking.

  Large-scale influential social events often have great educational effects. Students extract useful geographic knowledge from geographic statistics and social news, process it, and integrate it with their current geographical knowledge. Obviously, this incident had a huge influence on China. Taking it as an example, natural disaster-related discipline knowledge is injected into life safety education, which increases students' attention and significantly improves the effectiveness of learning.

  (2)     Contrast learning is an effective way of memorizing.

  The direct cause of flood disaster and mountain landslide is heavy rain, but there are differences; flood disasters are meteorological disasters, mountain landslides are geological disasters. By comparing the lists, students learn their concepts, generation mechanisms, and manifestations of disasters. Students can subtly distinguish them clearly and grasp the ontology more insightfully.

*5.3. Analyze the Climatic Conditions of Zhengzhou When the Rainstorm Occurred*

- Data source: Typhoon Network of the Central Meteorological Observatory
- Technical route:

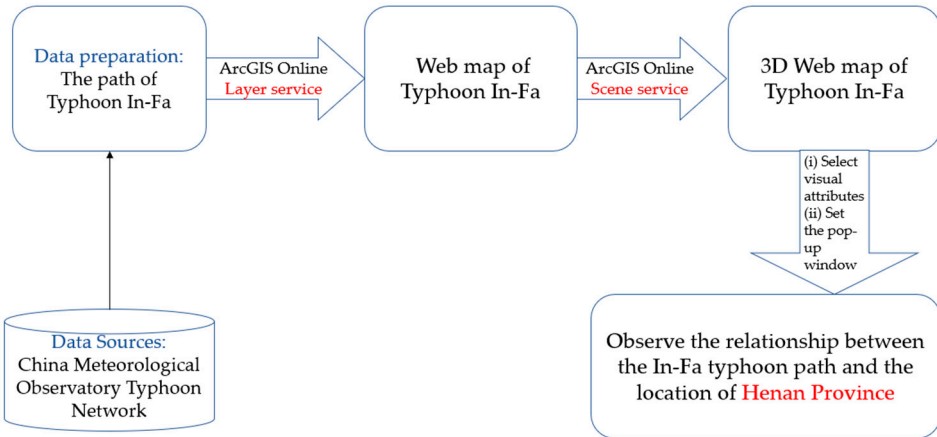

The typhoon "In-Fa" path data as of noon on 24 July 2021 was obtained, and Comma Separated Value text file schema (Table 2) were generated by sorting out the data. Web maps were generated in ArcGIS Online and feature layers were saved. The instructions were as fol-

lows: open the scene viewer, add the required feature layers (typhoon path, Chinese boundaries, provinces, etc.), select the main attributes to be visualized, select the drawing style "3D Count and Amounts", and finally generate the intuitive and vivid 3D scene (Figure 10) as shown in the picture (https://arcg.is/1z4Wer0, accessed on 20 February 2022).

**Table 2.** Comma Separated Value text file schema.

| Feature | Feature Description |
|---|---|
| Beijing Time | Take Beijing Time as the reference |
| Typhoon Class | Tropical storm/severe tropical storm/typhoon/strong typhoon |
| Longitude | Longitude position of typhoon center |
| Latitude | Latitudinal position of typhoon center |
| Central Air Pressure (hPA) | Pressure at the center of the typhoon, in hPA |
| Max. wind speed (m/s) | Maximum wind speed at the center of the typhoon in m/s |
| Wind direction | Wind direction at the center of the typhoon |

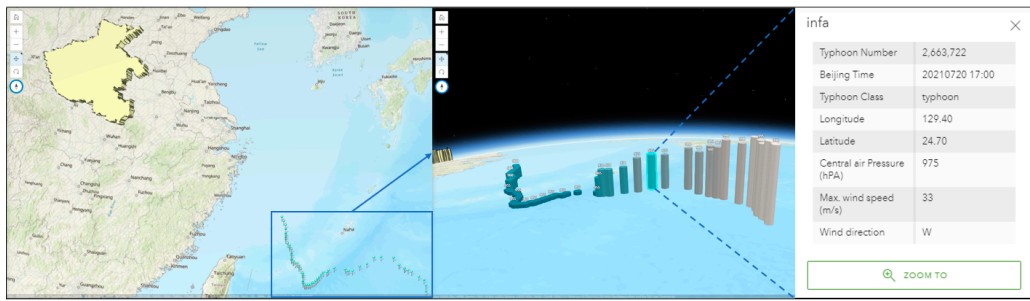

**Figure 10.** The moving path of typhoon "In-Fa" as of noon on 24 July 2021.

- Inquiry activity:
  There are three reasons for "7.20 Zhengzhou Torrential Rain". One is the typhoon "In-Fa", which cooperated closely with the subtropical high airflow, and a large amount of water vapor was transported from the sea to the mainland through easterly winds, thus producing a large amount of precipitation. Second, when the easterly wind blows to Henan, it meets Taihang Mountain and Funiu Mountain, and the mountain front converged and uplifted so that the scope of the rainstorm was locked in Henan again. Third, the atmospheric circulation situation was stable, increasing the duration of heavy rain.
  (1) How far away is the culprit of this rainstorm, "In-Fa", from us (Henan is located in the inland area)?
  (2) Why is the airflow of water vapor transported by typhoon "In-Fa" from an easterly direction?

- Demonstration operation:
  (1) First, select the column in the 3D scene at a specific time; then use the analysis-ranging tool to derive the center of the typhoon fireworks at this moment is nearly 2000 km away from Zhengzhou (Figure 11).
  (2) During the summer, the wind direction of Typhoon "In-Fa" was clockwise, the Northern Hemisphere Subtropical High cyclone center counter-clockwise (Figure 12). The interaction between the typhoon "In-Fa" and the Subtropical high caused the warm and wet air from the Pacific to move eastward, directly to Henan.

- Design Intention: Improve the spatial perception of high school students.
  Due to the limitations of its naturalness and regional characteristics, it is difficult to use geography to make natural phenomena on-site and real. 3D Mapping is a convenient way to assist students to create a profound impression of a particular area

and the relative position between different areas. Typhoons are unfamiliar to high school students living in mainland China. Because of the complexity of the Earth's environment and the close connection between ecological environments, geography teachers are required to choose the theme of natural disaster education according to local conditions. Contrast learning is an efficient way of memorizing.

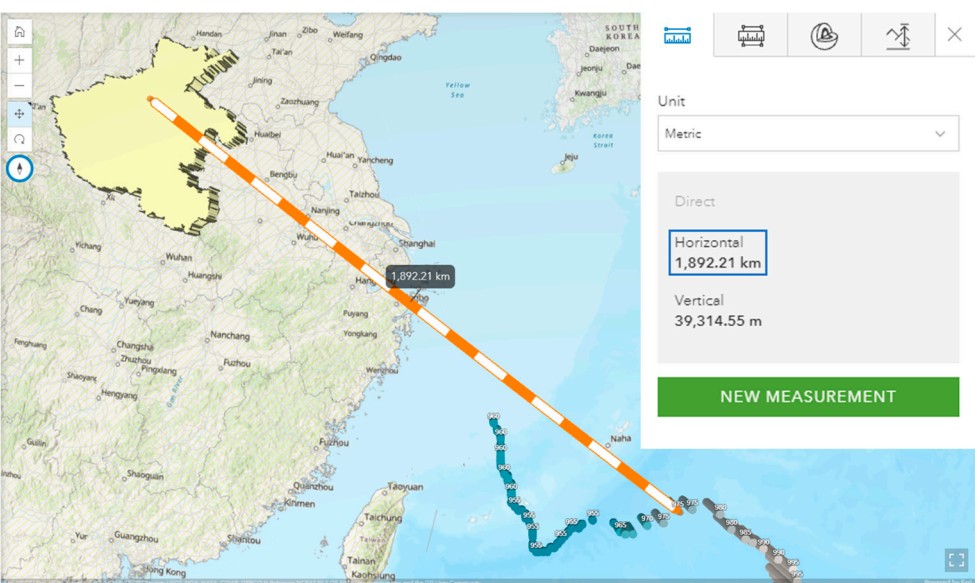

**Figure 11.** Analysis-ranging tool in ArcGIS Online 3D scene.

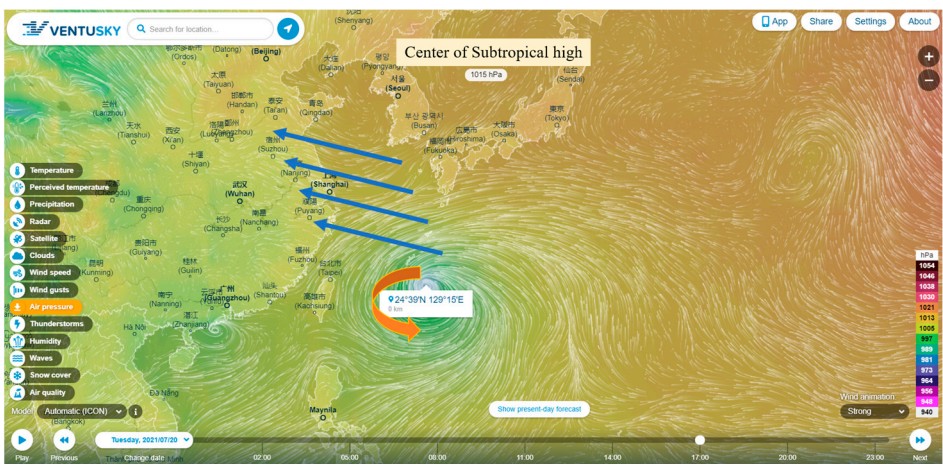

**Figure 12.** Real-time pressure map of VENTUSKY at 5 p.m. 20 July 2021.

*5.4. The Role of Geographic Information Technology in Natural Disaster Education*

Students observed the formation process of flood and typhoon disasters through the Internet (China Meteorological Network, Zhengzhou Meteorological Bureau, etc.). For disaster rescue, Gaode Map released a real-time rainstorm mutual aid layer. The trapped people only need to click the "water" button, which can automatically obtain the real-time location and send road condition information, so that firefighters can carry out rescue at the fastest speed.

After the rainstorm fades, firefighters and sanitation workers need to check and repair urban public facilities in Zhengzhou, and then they can use the Quick Capture for ArcGIS (Mobile APP) to collect information on the state of urban public facilities after the rainstorm (Figure 13). Students can also serve as volunteers to collect information on the condition of public facilities in the city after a rainstorm. The minimalist user interface makes it easy to quickly capture field data such as location, field conditions, and

even photos without interrupting the work at hand. Meanwhile, the data will be sent back to relevant government departments in real-time for analysis. Major maintenance workers will be selected for particular areas to increase the restoration efficiency of the city. (https://arcg.is/yv4rT0, accessed on 20 February 2022)

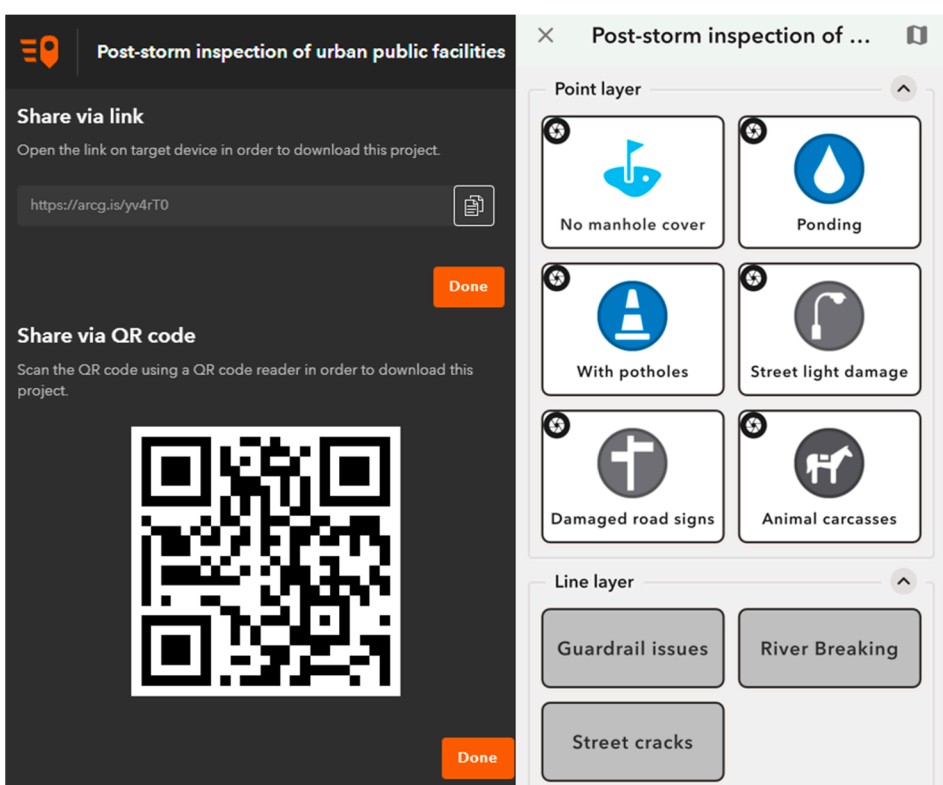

**Figure 13.** Quick Capture's information collection and sharing interface.

## 6. Discussion

The inquiry-based teaching case of natural disaster education in the above case can be used as a reference for other educators in their research. It is not immutable. Educators can proceed most appropriately according to surrounding environmental conditions and relevant social events. In the process of making smart maps and story maps, teachers can choose different templates according to their own needs and are not limited to those provided in the case. It is better to select cases involving different aspects and different levels. For example, the cases selected in this paper involve many natural disasters such as a rainstorm, typhoon, mountain landslide, and urban waterlogging to cultivate students' ability to find and solve problems independently.

Education in the 21st century is a cornerstone of society and a key engine of national success, and 21st-century students need 21st-century skills [41]. On the definition and contents of the 21st-century trend of diversity, its essence is powerful communication and collaboration skills, technical expertise, innovative and creative thinking skills, and the ability to solve problems [42].

Web GIS brings opportunities and challenges to the whole educational field. Maria de Lázaro Torres et al. [43] pointed out that Web GIS is an indispensable supplementary element to improve the quality of education and teaching, so that teaching activities can be carried out in an orderly manner and students can learn meaningfully. With its unique interaction, Web GIS can help students improve their ability of collaboration and communication.

Web GIS can be used in a wide range of fields. In addition to the natural disaster education mentioned in this paper, urban landscape, topography, and geomorphology of a certain region and regional geography in fundamental geography courses can be

well integrated with a particular function of Web GIS. The use of story maps has become quite widespread, and many schools in the United States are equipped with unique story collections of maps, which shows that the prospects for Web GIS are very positive.

## 7. Conclusions

Natural disaster education is not accomplished immediately in high school, it is progressively learned and mastered by students in the process of growing up. Natural disaster education requires multi-disciplinary teachers at different grades to educate students about natural disasters with the corresponding curriculum. School disaster education is essential to raise awareness among students and encourage preparedness action. Countries around the world are taking measures to improve programs of disaster education. For example, in Australia, school disaster education for children is actively promoted [44]. The study identified a significant imbalance between knowledge and skills as a prominent problem in the natural disaster education process. Therefore, how to improve students' disaster prevention skills is a priority for educators in the future. Natural disaster education is not only the responsibility of the geography curriculum; each discipline has its different division of labor. Breaking disciplinary boundaries and integrating disaster education with multi-disciplinary ideas will enhance the learning effectiveness of disaster education and play its full active function.

Teachers have a big role in developing good character in the information era [45]. Information technology is growing rapidly, more and more Web GIS platforms are being established. In addition, a large number of update online geographic databases also provide convenience for geography teaching, mapping, and curriculum resource development, such as Google Earth Engine [46,47]. Geography teachers can use more Web GIS tools in the process of information teaching and research. According to the school's hardware and software basis and different teaching courses, teachers choose the most appropriate functions and teaching tools. Due to the lagging economic development, some developing countries lack systematic teacher training related to information technology and the information literacy of geography teachers is poor. Web GIS does not require complicated software installation steps, and teachers can use it to carry out teaching and research activities. This paper also aims to help geography teachers overcome the information barrier.

Compared with the traditional natural disaster education model, the Web GIS inquiry-based teaching case pays more attention to the classroom interaction between teachers and students, the updating of teaching content and materials, and the cultivation of students' inquiry and practice ability. The application of geographic information systems in school geography has entered a new era. In this new era, it is very promising to support geography teaching and learning by using Web GIS for high-quality mapping, spatial analysis, and geographic visualization.

It is worth emphasizing that this research has laid a foundation for Web GIS to better guide the teaching of geography and emphasized the importance of natural disaster education in a Geography Fundamentals Course. In the future, research on the integration of Web GIS and geography education will continue to help geography teachers break information barriers and promote the professional development of information technology for geography teachers. These are all effective explorations of geography education in the information era.

**Author Contributions:** This research was carried out with the cooperation of all authors. Conceptualization, H.X. and Y.Q.; methodology, J.L.; data curation, X.Z.; writing—original draft preparation, J.L.; writing—review and editing, H.X., J.L. and X.G.; visualization, R.L.; supervision, H.X. and P.F.; project administration, Y.Q. All authors have read and agreed to the published version of the manuscript.

**Funding:** This research was funded by Key project of Higher Education teaching Reform research and Practice in Henan Province (2019SJGLX043); Key project of Undergraduate Teaching Reform research and Practice of Henan University (HDXJJG2018-01, HDXJJG2019-48); Teacher teaching Development project of Henan University's School-level Education Reform Project (YB-JFZX-09); and Environment

and Planning Teaching reform project of national Experimental Teaching Demonstration Center (2020HGSYJX007).

**Institutional Review Board Statement:** Not applicable.

**Informed Consent Statement:** Not applicable.

**Data Availability Statement:** "Typhoon Network of the Central Meteorological Observatory" at http://typhoon.nmc.cn/web.html (accessed on 20 February 2022).

**Acknowledgments:** We thank anonymous reviewers for their valuable comments and suggestion.

**Conflicts of Interest:** The authors declare no conflict of interest.

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
