# Peer review of "Web GIS for Sustainable Education: Towards Natural Disaster Education for High School Students"

_sustainability, doi:10.3390/su14052694_

Round 1
Reviewer 1 Report
Authors must revise the discussion section, there was need to improve discussion about education and 21st century skill.
Reviewer 2 Report
Sustainability – ID: 1580342, article review
Dear authors,
your paper Web GIS for Sustainable Education: Towards Natural Disaster Education for High school students deals with a very current topic of concrete use of Web GIS in geography teaching. The article is well structured and informative.
Suggestions for improving the paper
The paper is well designed and for improving the content I would suggest to the authors to expand the list of natural catastrophes with the following examples
- wildfires;
- human, animal and plant diseases;
- plagues (like grasshopper events)
- turbidites
- tsunamis
- seiches
- tornados
- avalanches
- strong winds
- volcanic eruptions
- soil collapses
Before finishing the work, consult the article:
Healy, G., & Walshe, N. (2020). Real-world geographers and geography students using GIS: Relevance, everyday applications and the development of geographical knowledge. International Research in Geographical and Environmental Education, 29(2), 178-196.
Kind regards
Reviewer 3 Report
The use of GIS methods in teaching in secondary schools has been a subject undertaken by numerous authors around the world for many years, which, unfortunately, cannot be seen in the text. Certainly, however, the increase in the availability of specialized software contributed to the popularization of the use of digital teaching methods in schools. The examples presented in the article are of course useful, however, they do not seem to be optimal for every country (examples should be as local as possible). Regardless of these comments, I believe that the article is an important contribution to the popularization of GIS methods in training education and provides useful tips for teachers. In addition, I believe that it should be possible to use alternative, free programmes, such as QGIS, which make it possible to use them in poorer countries.
Reviewer 4 Report
General comment
your article is Quite interesting, Unfortunately, not well-elaborated. improvements need to be made to sharpen the substance of the article
Abstract
1. The abstract should be written in good English and should include the aim, general method, and significant results
2. Mention how the stages of the research method process are especially using GIS in education in China
Introduction and Theoretical Background
- The introduction has not clearly conveyed the problem and the theory referred to so that there is a missing link
- Need more prior research’s to strengthen the problem statement and state-of-the-art
- There needs to be an elaboration of previous studies that show in detail the renewability of this research
- The background needs to be more detailed elaborated, including problems at the research location, gaps, and previous research that has been done.
- how so far the use of GIS in the education system in China needs to be elaborated
Method
- There should be an explanation of key variables that have been explained and included in the conceptual framework so that it is clear what relationship between the variables will be tested for GIS, education, and disaster
- The method does not discuss the theory regarding the method used again, please focus more on the application process and the method used
Results and Discussions
- There is no explanation of how tables and figures are elaborated so that it becomes a conclusion there needs to be an explanation of how the technicality of generating data. it may also be appropriate to include the appropriate source citations
- At the stage of the results and discussion are no longer discussing the theory, but rather the emphasis of discussion with the stages of processing data with methods and show the results for GIS, education, and disaster
- There is no explanation of how tables and figures are elaborated so that it becomes a conclusion there needs to be an explanation of how the technicality of generating data.
- At the stage of the results and discussion are no longer discussing the theory, but rather the emphasis of discussion with the stages of processing data with methods and show the results
Conclusion
- The conclusion must go straight to the core of the matter. Is the background problem solved? I don't think so
- The conclusion reads more like an extension of the discussion. The conclusion is that the initial tests give evidence that the proposed treatment merits further investigation
- References have not shown a good state of the art it should be updated so that the research novelties are visible
Reviewer 5 Report
Los cambios necesarios son:
- Incrementar la teoría sobre la inclusión de las tecnologías en los procesos educativos
- Incluir conclusiones sustentadas en bibliografía científica de estudios actuales
-Revisión APA
Round 2
Reviewer 4 Report
Nice Improvement
but please describe of novelty this article in introduction and conclusion
